# Enhancing the Quality of Indoor-Grown Basil Microgreens with Low-Dose UV-B or UV-C Light Supplementation

**DOI:** 10.3390/ijms26052352

**Published:** 2025-03-06

**Authors:** Ernest Skowron, Magdalena Trojak, Ilona Pacak, Paulina Węzigowska, Julia Szymkiewicz

**Affiliations:** 1Department of Environmental Biology, Jan Kochanowski University of Kielce, Uniwersytecka 7, 25-406 Kielce, Poland; magdalena.trojak@ujk.edu.pl (M.T.); paulina.wezigowska@gmail.com (P.W.); julia.szymkiewicz@ujk.edu.pl (J.S.); 2Department of Medical Biology, Jan Kochanowski University of Kielce, Uniwersytecka 7, 25-406 Kielce, Poland; ilona.pacak@gmail.com

**Keywords:** basil, indoor farming, spectrum supplementation, ultraviolet light, plant secondary metabolism, UVR8 receptor, chalcone synthase

## Abstract

Controlled-environment crop production often weakens plants’ defense mechanisms, reducing the accumulation of protective phytochemicals essential to human health. Our previous studies demonstrated that short-term supplementation of low-dose ultraviolet (UV) light to the red–green–blue (RGB) spectrum effectively boosts secondary metabolite (SM) synthesis and antioxidant capacity in lettuce. This study explored whether similar effects occur in basil cultivars by supplementing the RGB spectrum with ultraviolet B (UV-B, 311 nm) or ultraviolet C (UV-C, 254 nm) light shortly before harvest. Molecular analyses focused on UV-induced polyphenol synthesis, particularly chalcone synthase (CHS) level, and UV light perception via the UVR8 receptor. The impact of high-energy UV radiation on the photosynthetic apparatus (PA) was also monitored. The results showed that UV-B supplementation did not harm the PA, while UV-C significantly impaired photosynthesis and restricted plant growth and biomass accumulation. In green-leaf (Sweet Large, SL) basil, UV-B enhanced total antioxidant capacity (TAC), increasing polyphenolic secondary metabolites and ascorbic acid (AsA) levels. UV-C also stimulated phenolic compound accumulation in SL basil but had no positive effects in the purple-leaf (Dark Opal, DO) cultivar. Interestingly, while the UV-B treatment promoted UVR8 monomerization in both cultivars, the enhanced CHS level and concomitant SM synthesis were noted only for SL basil. In addition, UV-C also induced CHS activity and SM synthesis in SL basil but clearly in a UVR8-independeted manner. These findings underscore the potential of UV light supplementation for enhancing plant functional properties, highlighting species- and cultivar-specific effects without compromising photosynthetic performance.

## 1. Introduction

Open-field farming faces significant challenges, including climate change and the decreasing availability of arable land and freshwater [1]. Consequently, there is increasing interest in controlled-environment agriculture, such as vertical or indoor farming (IF) that can produce food in a climate-resilient manner with lower land and water use [2]. Moreover, IF offers several advantages compared to conventional agriculture, including pest and drought mitigation, year-round production, and reduced transportation costs [1,3]. On the other hand, plants grown inside such facilities have limited exposition to natural sunlight components, as the artificial lighting spectrum is mostly depleted of UV radiation [4]. Consequently, IF-grown plants are deprived of natural stress stimuli, which are crucial for the synthesis and accumulation of secondary metabolites (SMs) [5]. Thus, in recent years, increasing attention has been paid to optimizing the spectrum composition of artificial light systems to enhance crop growth and quality [6]. In particular, analyses of low-fluence UV light supplementation are worth considering due to its effect on photomorphogenesis and nutritional properties of plants [7,8].

Plant responses to UV light involve specific photoreceptors, such as cryptochromes (CRYs), which respond to blue (400–500 nm) and UV-A light (315–400 nm), or UV Resistance Locus 8 (UVR8), which responds to UV-B (280–315 nm) [9]. However, the effect of UV-C radiation (100–280 nm) has been significantly less studied because of its high absorption by the ozone layer, restricting penetration to the earth’s surface. However, the widely accepted mechanism underlying plant responses to this spectrum region is linked to elevated reactive oxygen species (ROS) production [9].

In the case of the UVR8 photoreceptor, it exists as an inactive dimer localized in the cytosol, but after receiving UV-B stimulation, the UVR8 homodimer dissociates into active monomers and subsequently enters the nucleus [10]. Interestingly, CRY- and UVR8-depended signaling pathways show similarities, as active homodimers of CRY, as well as the monomers of UVR8, interact with the E3 ubiquitin ligase CONSTITUTIVELY PHOTOMORPHOGENIC 1 (COP1) preventing ELONGATED HYPOCOTYL 5 (HY5) transcription factor ubiquitination and degradation and in turn activates downstream gene expression [11]. HY5 is a key positive regulator of light signaling and photomorphogenesis, promoting SM synthesis, particularly phenolic compounds due to the induction of chalcone synthase (CHS) gene expression [12]. The negative regulators of the UVR8-COP1-HY5 pathway are REPRESSOR OF UV-B PHOTOMORPHOGENESIS, RUP1/RUP2 proteins, inducing UVR8 dimerization back to an inactive form, or B-Box transcription factors (i.e., BBX24 protein) that suppress HY5 activity [13]. Consequently, UV treatments are increasingly recognized as effective tools for biofortifying crops with functional phytochemicals and enhancing antioxidant capacity. Among the bioactive compounds, flavonoids are the most abundant group, including flavanones, flavonols, anthocyanins, and others [14].

Our previous research demonstrated that UV-A exposure increases the total phenolic (TPC) and flavonoid content (TFC) in lettuce and basil, especially in green-leaf cultivars [15]. Similarly, UV-B exposure enhances TPC, TFC, anthocyanins (ANT), carotenoids, and ascorbic acid (AsA) in lettuce [16]. In this study, we evaluated the effect of short-term supplementation of the artificial red–green–blue (RGB) LED spectrum with low-dose UV-B (311 nm, cumulative dose—CD of 15.622 kJ m^−2^) and UV-C (254 nm, CD of 6.008 kJ m^−2^) in two basil (*Ocimum basilicum* L.) cultivars: Sweet Large (SL, green leaves) and Dark Opal (DO, purple leaves). Basil is a highly valued medicinal and culinary crop, renowned for its distinctive aroma, flavor, and rich bioactive compounds. Notably, the composition of the light spectrum plays a crucial role in regulating the phenolic and flavonoid content of basil leaves. Moreover, studies have demonstrated the differential responses and spectrum requirements among green- and purple-leaf basil cultivars when grown under distinct light conditions [17,18].

In this study, effects of UV exposure were assessed through photosynthetic activity (chlorophyll *a* fluorescence), photosynthetic pigments, protein levels, lipid peroxidation rate, and the level of the following SMs: TPC, TFC, ANT and AsA. Moreover, we also analyzed the relationship between UV-B or UV-C spectrum supplementation, UVR8 activation, and the abundance of CHS protein. The results show that UV-B exposure significantly increased TFC, ANT, and AsA levels in the green-leaf basil and enhanced AsA content in the purple-leaf cultivar without an adverse impact on photosynthetic activity or plant morphology. Conversely, UV-C exposure reduced photochemical efficiency in the green-leaf basil and exerted a negative impact on the morphology, growth, and leaf biomass of both cultivars. While UV-C light was less effective than UV-B in promoting TFC and AsA accumulation in the green-leaf basil, it significantly increased TPC and ANT levels. In the case of the purple-leaf basil upon UV-C treatment, no further SMs or AsA accumulation was noted above the high initial level. On the molecular level, the results showed that the UV-B, but not the UV-C, light induced UVR8 monomerization in both cultivars. At the same time, UV-B/-C exposure induced CHS level above the control level in only in the green-leaf cultivar and consequently enhanced phenolic compound accumulation.

This study offers insight into the potential of UV-B and UV-C supplementation in standard RGB lighting systems, which typically lack UV components, to improve the quality of plant products due to the increased level of antioxidants. Secondly, it provides evidence of a positive relationship between UV-B perception and UVR8 activation but not necessarily with an enhanced CHS level and polyphenol biosynthesis.

## 2. Results

### 2.1. Effects of UV Light on Plant Morphology and Biomass Accumulation

Interestingly, we noted that both cultivars at the end of the UV-B light treatment showed some growth stimulation compared to plants grown solely under the RGB regime (Figure 1a,b and Figure 2a,b). On the other hand, UV-C exposure caused severe visible morphological changes in both cultivars, including growth inhibition and necrotic, bleaching lesions along leaf edges, leaf glazing, and curling (Figure 1c and Figure 2c).

Consequently, both cultivars grown under the RGB spectrum supplemented with UV-B light showed increased leaf biomass accumulation (Figure 3). We noticed especially that UV-B enhanced the fresh mass (FM) by 52% and 32% and the dry mass (DM) by 49% and 43% in SL (Figure 3a) and DO (Figure 3b), respectively. The opposite reaction was observed for the UV-C treatment, as its application decreased FM by 29% and 35%, while DM was decreased by 25% and 29% in the leaves of SL and DO, respectively.

### 2.2. Photosynthetic Activity Under Short-Term Exposition to UV-B or UV-C Light

#### 2.2.1. Effects of UV Light on Photosynthetic Pigments and Soluble Leaf Protein Content

Both cultivars exhibited similar chlorophyll *a+b* and carotenoid levels under the RGB spectrum (Table 1 and Table 2). UV-B exposure reduced chlorophyll *a + b* by 38% in SL and 43% in DO. In contrast, UV-C had no significant effect. In DO, UV-B reduced carotenoids by 51%, corresponding to a 44% and 41% decline in chlorophyll *a* and *b*, respectively (Table 2). In SL, UV-B reduced chlorophyll *a* and *b* by 38% and 36%, respectively, along with a 30% carotenoid reduction. Upon UV-B treatment, the chlorophyll *a*/*b* ratio decreased by approximately 5% in both cultivars (Table 1 and Table 2). UV-C exposure slightly increased carotenoids in DO by 4%.

Soluble leaf protein (SLP) content also varied between the treatments. In SL, UV-B and UV-C increased SLP by 24% and 29%, respectively. In contrast, in DO, SLP levels decreased by 17% and 20% following UV-B and UV-C exposure. Interestingly, under RGB, SLP in SL was over 50% lower than in DO (Table 1 and Table 2).

#### 2.2.2. Effects of UV Light on RuBisCO Abundance

Electrophoretic separation and densitometric analysis revealed that the UV treatment influenced RuBisCO accumulation. In SL, UV-B light reduced the large subunit (LSU) by 16%, while the small subunit (SSU) increased nearly 2-fold. Under UV-C, LSU decreased by 28%, whereas a 20% increase in the SSU level was noted (Figure 4a,c). In contrast, in the DO cultivar upon UV-B treatment, the LSU and SSU abundance increased by 42% and 17%, respectively, while UV-C increased only the LSU level by 38% (Figure 4b,c).

#### 2.2.3. Effects of UV Light on Photosynthetic Efficiency of PSII

Chlorophyll *a* fluorescence kinetics were used to assess photosynthetic activity. UV-B did not affect the maximum quantum efficiency of PSII (Fv/Fm) in either cultivar (Figure 5a and Figure 6a). However, in SL upon the UV-B treatment, an effective quantum yield of PSII photochemistry (ΦPSII) and electron transport rate (ETR) (Figure 5b,f) slightly decreased by day 3, accompanied by increased non-photochemical quenching (ΦNPQ) (Figure 5c). Still, however, the noted decreases were statistically insignificant, and by day 4, the mentioned parameters returned to control levels, indicating light acclimation. In the DO cultivar, UV-B light induced an initial increase in regulated and decrease in non-regulated energy dissipation (Figure 6d,e). In contrast, UV-C exposure impaired photosynthetic activity from the first day, especially in the SL cultivar, noted with reduced ΦPSII and ETR, and there was a concomitant increase in the non-photochemical energy distribution (Figure 5d). Moreover, while the DO cultivar acclimated after day 3 (Figure 6b,f), the SL cultivar showed prolonged reductions in photochemical efficiency and heat dissipation capacity.

### 2.3. Antioxidant Capacity in Response to Supplemental UV-B or UV-C Light

#### 2.3.1. Total Phenolic and Flavonoid Content

The total phenolic content (TPC) was expressed as µg of gallic acid equivalents per mg of fresh weight (FW) (Figure 7). Green-leaf basil exhibited a 14% lower TPC under the control RGB spectrum (Figure 7a) compared to purple-leaf basil (Figure 7b). Short-term UV-B supplementation did not affect TPC in the Sweet Large (SL) cultivar but resulted in a 4% reduction in the Dark Opal (DO) cultivar. In contrast, UV-C exposure increased TPC in SL by 9% while decreasing it by 6% in DO.

In the case of the total flavonoid content (TFC), it was expressed as µg of rutin equivalents per mg of FW (Figure 8). The DO cultivar exhibited significantly higher TFC than SL, with levels under the control RGB spectrum being at least 2.8 times higher in DO. UV-B exposure led to a 3.5-fold increase in TFC in SL (Figure 8a) but caused a 30% decrease in DO (Figure 8b). UV-C treatment reduced TFC in SL by 16% (Figure 8a) but had no significant effect in DO (Figure 8b).

#### 2.3.2. Anthocyanin Level

Anthocyanins (ANTs) were significantly more abundant in the purple-leaf DO cultivar, with levels exceeding those in SL by over 60-fold under the control spectrum (Figure 9). UV-B exposure increased ANT accumulation in SL by 15% (Figure 9a) but did not reach DO levels. In DO, UV-B caused a 13% reduction in ANT (Figure 9b). Upon UV-C treatment, ANT accumulation in SL increased by 23% but did not change in DO.

#### 2.3.3. Ascorbic Acid Pool

Ascorbic acid (AsA) levels were analyzed in both reduced (AsA) and total (AsA + dehydroascorbic acid, DAsA) forms. In SL, most AsA remained in the reduced form under the RGB spectrum, except after UV-B and UV-C exposure, which increased total AsA by 2.6- and 1.4-fold, respectively (Figure 10a). The DO cultivar exhibited a total AsA pool more than 30 times higher than SL under RGB (Figure 10b). However, unlike in SL, the UV-B and UV-C did not alter the total AsA-to-initial-AsA ratio in DO (Figure 10d). Still, UV-B increased AsA levels in DO by approximately 25%. Overall, the results show that SL was more sensitive to UV-B light than DO, as indicated by an increase in both initial and total AsA (Figure 10a) and an elevated AsA+DAsA/AsA ratio (Figure 10c).

#### 2.3.4. Overall Antioxidant Capacity

Total antioxidant capacity (TAC) was measured as µg of butylated hydroxytoluene (BHT) equivalents per mg FW (Figure 11a,b), while DPPH (2,2-Diphenyl-1-picrylhydrazyl) radical scavenging activity was assessed using a BHT calibration curve (Figure 11c,d). Under the RGB spectrum, SL basil exhibited a 6.5 times lower TAC than DO. UV-B exposure increased TAC in SL by 6-fold (Figure 11a) alongside a 129% increase in radical scavenging activity (Figure 11c). In DO, despite a rise in AsA levels, the UV-B treatment resulted in decreased TAC due to lower TPC, TFC, and ANT, while DPPH radical scavenging remained unchanged at 79% (Figure 11d). UV-C exposure had no significant effect on TAC or scavenging activity in either cultivar (Figure 11b,d).

Pearson’s correlation analyses revealed that in SL, TAC was strongly (*p* ≤ 0.001) linked to TFC and AsA levels (Figure 12a), whereas in DO, TAC correlated with TPC, TFC (*p* ≤ 0.001), and ANT (*p* ≤ 0.001) (Figure 12b). These findings indicate that UV-induced bioactive compound synthesis enhanced the antioxidant properties in both basil cultivars but via distinct mechanisms in the green- and purple-leaf plants.

#### 2.3.5. Effects of UV Light on Lipid Peroxidation

Lipid peroxidation, measured via TBARS assays, indicated oxidative stress responses. In SL, UV-C exposure increased TBARS levels by 48% (Figure 13a), while UV-B reduced them by 28%. In DO, UV-B and UV-C reduced TBARS levels by 47% and 29%, respectively (Figure 13b). Notably, DO exhibited 1.2 times higher TBARS levels under RGB than SL.

#### 2.3.6. Immunoblot Analyses of UVR8 and CHS Protein

The results showed that UV-B and the UV-C treatment increased CHS levels in the SL cultivar by approximately 8% and 12%, respectively (Figure 14a,b). However, in the case of DO basil, the CHS level was diminished after UV-B treatment by 20% (Figure 14a,c). At the same time, UV-B light promoted UVR8 dimer dissociation, forming a monomer, the level of which increased in SL and DO by 69% and 53%, accompanied by decreased UVR8 homodimer levels by nearly 20% and 41%, respectively (Figure 14a,d,e). Upon UV-C photon perception, no UVR8 activation was observed, but the overall UVR8 dimer level was lowered compared to the control by 8% in the SL cultivar and 15% in the DO cultivar. Interestingly, low levels of UVR8 monomer were also detected in both the RGB and RGB+UV-C plants.

## 3. Discussion

### 3.1. Efficiency of RGB Spectrum Supplementation with UV-B or UV-C Light on Antioxidant Capacity, UVR8 and CHS Activation, and Morphology of Basil Cultivars

The consumption of fresh plant products is strongly associated with disease prevention due to the antioxidant activity of their secondary metabolites [19]. However, in controlled-environmental agriculture, the synthesis of bioactive compounds is often limited by the absence of abiotic stress, such as UV radiation [20]. For this reason, this study investigated the impact of low-dose UV-B or UV-C treatment applied before harvest on two basil cultivars: green- (SL) and purple-leaf (DO), cultivated as microgreens. The first goal was to enhance antioxidant properties without negatively impacting photosynthetic efficiency and plant morphology. The second goal was to investigate and compare the mechanism underlying UV-induced polyphenol synthesis in plant tissue due to UVR8 activation and a concomitant increase in CHS activity in the cultivars presenting varied basic levels of SM.

Our previous studies proved the efficacy of preharvest UV-A and UV-B light in increasing antioxidant levels in leafy greens such as lettuce and basil [15,16]. The observed effect of UV-B light is related to enhanced flavonoid biosynthesis via the upregulation of key enzymes of the phenylpropanoid and flavonoid biosynthetic pathways, such as CHS and phenylalanine ammonia-lyase (PAL) [21]. Another study [22] analyzing purple-leaf basil (cv. Red Ruby) documented that UV-C light was even more effective in increasing ANT levels compared to UV-B. At the same time, however, authors [23] documented that green basil (cv. Genovese) exposed to 1–2 h day^−1^ of UV-B presented decreased both TPC and TFC.

In this study, a four-day progressive UV-B supplementation to the RGB background, applied directly before harvest effectively increased total TFC and ANT in the SL cultivar. Conversely, in DO basil, UV-B reduced all analyzed phenolics due to the high baseline levels of UV-screening pigments, which attenuate CHS activation and concomitant phenolics synthesis, despite efficient UVR8 monomerization (Figure 14). Interestingly, in our previous study [16], we also documented that green lettuce showed a greater response to UV-B than a reddish one; however, UV-B light was still able to increase the phenolic content in both lettuce cultivars. In the case of AsA, UV-B exposure was also more effective in increasing its level in the green cultivar than in the purple cultivar, noted with a higher level of total AsA, which was present mostly in the oxidized form. Like the previous study [24], the green-leaf basil cultivars showed reduced ascorbate levels after UV-B exposure.

Overall, in our study, the UV-B treatment boosted the total antioxidant capacity (TAC) of the green basil extracts by approximately five times while lowering its value in the purple-leaf plants. A possible explanation for these results is a significantly higher initial accumulation of phenolics and AsA in purple basil, acting as a screening shield against UV-B (and partially for UV-C). Lower antioxidant levels in green cultivars make them more susceptible to UV-induced synthesis of SM. Consequently, the high initial level of screening pigments has been earlier postulated [24] as a key factor contributing to a slower accumulation of flavonoids. However, as the results showed that the UV-B treatment was efficient in activating UVR8 monomerization in both cultivars, the downstream signaling in the DO cultivar was thus impaired. A likely explanation for the limited further accumulation of SMs in DO is the initially heightened activity of HY5, which functions upstream of polyphenol structural enzymes [25] but also upregulates RUP1 and RUP2, which serve as essential negative feedback regulators by promoting UVR8 dimerization, thereby facilitating its reversion to the ground state [26]. Interestingly, the analyzed DO cultivar, besides possessing a high concentration of ANT in its tissue, also presented restricted growth. Both of these traits have been also identified with pear, strawberry, Arabidopsis, tobacco, and tomato and are related to allelic variation in the BBX24 transcription factor [25]. In such a scenario, plants with altered BBX24 function preserve UV-B sensitivity, yet the HY5 activity is enhanced as the BBX is no longer a negative regulator of the UV-B signaling pathway [10].

Additionally, a previous study [27] provides evidence of strong competition between flavonol and anthocyanin biosynthesis in acyanic- (green) and cyanic-leaf (purple) cultivars. The authors conclude that light attenuation by epidermal ANT in dark-pigmented cultivars represses the biosynthesis of colorless flavonoids, as their synthesis originates from the very same intermediate substrates, i.e., dihydro-flavonols. In fact, our results proved the negative correlation between TFC and ANT, but only in SL basil (Figure 12a,b). Moreover, the UV-B but not UV-C light exposure targeted preferentially TFC synthesis rather than ANT in SL. Such competition was not observed for the DO cultivar, which presented a high initial level of ANT.

UV-C exposure (254 nm peak) elicited an influence on the antioxidant composition distinct from UV-B exposure. In SL basil, UV-C significantly increased TPC, ANT, and AsA levels while reducing the TFC and carotenoid contents, resulting in no net change in TAC. In DO basil, UV-C only slightly increased the carotenoid pool and simultaneously decreased TPC and AsA. Consequently, upon UV-C exposure, a reduction in TAC was noted. Also, our previous research [16] showed that UV-C light exerted a negative impact on the analyzed phenolic compounds and carotenoid content in reddish-leaf lettuce. These findings and previous research underscore the cultivar-specific effects of UV-C and its potential to induce phytochemical synthesis, which might be more effective when applied in in vitro cultures [28,29].

The differential effects of UV-B and UV-C on antioxidant synthesis are attributed to their distinct modes of action. UV-B promotes flavonoid biosynthesis via UVR8-HY5 signaling, which has been proven with UVR8 receptor analyses (Figure 14), whereas UV-C primarily induces reactive oxygen species (ROS) generation, modulating phenolic metabolism through the mitochondrial electron transport chain and NADPH oxidase, affecting phenolic biosynthesis [30]. Consequently, the SL cultivar that showed a higher UV responsiveness for SM synthesis was also the cultivar that presented a higher level of lipid peroxidation rate assessed with TBARS (Figure 13a,b). Still, however, UV-C exposure caused morphological damage (e.g., leaf glazing and necrotic spots) (Figure 1c and Figure 2c) and significantly reduced FM and DM in the leaves in both cultivars (Figure 3), reducing plant quality.

### 3.2. Condition of Photosynthetic Apparatus in Response to UV-B or UV-C Supplementation to the RGB Spectrum

Ultraviolet radiation, particularly UV-B and UV-C, is well documented to adversely affect the photosynthetic apparatus (PA) [31]. UV-B exposure impairs key components of photosynthesis, including plastoquinone function, RuBisCO activity, and ATP synthase, and it reduces chlorophyll and carotenoid levels [32]. However, low doses of UV-B can induce protective responses, such as flavonoid accumulation, mitigating damage to the PA [16].

In the present study, the UV-B treatment significantly decreased the chlorophyll and carotenoid levels in both basil cultivars (Table 1 and Table 2) but did not impair photosynthetic activity, as measured by chlorophyll fluorescence parameters (Figure 5 and Figure 6). This suggests that UV-B-driven polyphenol synthesis attenuates excessive light energy, preventing photodamage [27]. Similarly, in a previous study [33], low-dose UV-B treatment of *Schisandra chinensis* (Trucz.) did not cause any significant changes in photosynthetic properties until the 15th day of treatment. At the same time, the highest phenolic content in the leaves was noted after 7 days. Also, when applied within *Mesembryanthemum crystallinum* [34], UV-B was shown to stimulate the non-enzymatic antioxidant machinery and TFC and TPC levels.

By contrast, UV-C caused more severe disruptions to PA integrity, likely due to high-energy photon damage to thylakoid structures and ROS-induced oxidative stress [35]. In SL basil, UV-C exposure led to an immediate reduction in photosynthetic activity, whereas the DO basil exhibited greater resilience, likely due to higher baseline levels of UV-absorbing compounds [36] and antioxidants such as AsA. Based on these findings, we conclude that the DO cultivar is less susceptible to both the beneficial effects of UV-C exposure, such as secondary metabolite accumulation, and its detrimental effects, including photosynthetic apparatus impairment and lipid peroxidation (Figure 13).

Overall, these findings align with previous studies [16,20], demonstrating that UV-C induces phytochemical accumulation, especially in green leaf cultivars, but it compromises morphological and physiological integrity, limiting its utility as a preharvest treatment, even when progressively applied in low doses.

## 4. Materials and Methods

### 4.1. Plant Material, Growth Conditions, Light Treatment, and Leaf Biomass Analysis

Microgreens of basil (*Ocimum basilicum* L.) with green (cv. Sweet Large, SL) and purple leaves (cv. Dark Opal, DO) were grown in P9 containers (9 × 9 × 10 cm) placed in environmentally controlled growth chambers. Plants were cultivated for 20 days after sowing (20 DAS) under LED RhenacM12 lamps (PXM, Podleze, Poland), providing 200 µmol m^−2^ s^−1^ of the RGB spectrum (wavelengths: 661, 633, 520, and 434 nm), applied alone (control, Figure 15a) or supplemented with UV-B (311 nm, PL-S 9W/01/2P 1CT/6X10BOX, Philips Lighting, Eindhoven, The Netherlands, Figure 15b) or UV-C (254 nm, TUV PL-S 9W/2P 1CT/6X10BOX, Philips Lighting, Figure 15c) during the last four days before harvest, as detailed in Table 3.

The photoperiod was 16/8 h (day/night; day 6.00 a.m.–10.00 p.m.), the average air temperature was maintained at 23/20 °C (day/night), the relative air humidity was kept at 50–55%, and the plants were treated with 430 ± 10 µmol mol^−1^ of CO_2_. The containers with basil plants cultivated under the same light illumination were turned away twice a day. To avoid canopy shading and overlapping, five plants per square meter of the illuminated area were cultivated. A total of thirty basil plants (three repetitions with ten plants per light treatment) were grown with each kind of light composition.

For the fresh biomass (FM) analyses, leaves (with petioles) were cut off using a sharp scalpel and immediately weighed. For the dry biomass determination (DM), the collected material was dried at 105 °C for 24 h and then weighed. Ten plants per light treatment were randomly selected for each biomass determination. Leaf biomass, either FM or DM, relates to the mass of all the leaves of the individual plant (with petioles).

### 4.2. Photosynthetic Pigment Determination

Chlorophyll *a*, *b*, and carotenoids were extracted from 20 mg of leaf tissue in 1 mL of DMSO, incubated at 65 °C for 3 h, and centrifuged (10,000× *g*, 5 min). Absorbance was measured at 480, 649, and 665 nm [37]. Six replicates were analyzed for each treatment.

### 4.3. Soluble Protein Levels, RuBisCO Abundance, and Western Blot (WB) Analysis

Soluble proteins were extracted using an alkaline lysis buffer (0.1 M NaOH, 0.05 M EDTA, 2% SDS, 2% β-mercaptoethanol) [38]. The protein content was measured using a NanoDrop spectrophotometer at 280 nm. For RuBisCO analysis, the samples were run on 4–20% gradient TGX gels (Bio-Rad, Hercules, CA, USA) at 200 V for 20 min. Gels were stained with Coomassie Blue, and protein bands were quantified using ImageJ (ImageJ v.1.52, National Institutes of Health, Maryland, Bethesda, MD, USA).

For WB protein, extracts were separated on 4–20% TGX stain-free gels (Bio-Rad), analyzed briefly for protein integrity under a UV illuminator and transferred to nitrocellulose membranes (0.45 µm pore) via semi-dry electroblotting (1.5 mA/cm^2^, 15 min). Membranes were blocked with 5% milk and incubated with the following primary antibodies overnight at 4 °C: anti-CHS (chalcone synthase; AS12 2615; 1:1000), anti-UVR8 (ultraviolet-B receptor UVR8; AS22 4716; 1:1000), and anti-eEF1a (elongation factor 1-alpha, loading control; AS10 934; Agrisera, Vännäs, Sweden). Detection was performed using HRP-conjugated secondary antibodies (AS09 602, Agrisera) and DAB substrate. Protein bands were quantified using ImageJ, normalized to eEF1a, and expressed relative to RGB-treated plants [39]. Each light treatment was analyzed in triplicate.

### 4.4. Chlorophyll Fluorescence (ChF)

ChF kinetics were measured with a PAM fluorometer (Maxi IMAGING-PAM, Walz, Effeltrich, Germany). Minimal fluorescence (Fo) was recorded under modulated blue light (0.01 µmol m^−2^ s^−1^), and maximal fluorescence (Fm) was determined after a saturating pulse (2700 µmol m^−2^ s^−1^) on dark-adapted leaves (30 min). The maximum quantum yield of PSII (Fv/Fm), the actual photochemical efficiency of PSII (ΦPSII), the quantum yield of regulated energy dissipation in PSII (ΦNPQ) and non-regulated energy dissipation in PSII (ΦNO), non-photochemical energy quenching (NPQ), and the electron transport rate in PSII (ETR) were recorded daily following UV exposure.

### 4.5. Estimation of Total Phenolic Content (TPC) and Total Flavonoid Content (TFC)

Estimation of TPC was conducted as described earlier [40]. In brief, 100 mg of fresh-weight (FW) leaf tissue was placed in tubes with 1.0 mL of methanol and incubated for 30 min at 60 °C. Then, the sample mixture was centrifuged at 10,000× *g* for 2 min. A total of 100 µL of each extract was mixed with 200 µL of 10% (*v*/*v*) Folin–Ciocalteu reagent (F–C) and vortexed twice for 10 s. Then, 800 µL of 700 mM Na_2_CO_3_ was added, vortexed twice for 10 s, and incubated for the next 30 min at 40 °C. After incubation, the mixture was centrifuged at 10,000× *g* for 1 min. For TPC determination, the absorbance (Abs) at 765 nm was estimated with a microplate spectrophotometer (Mobi, MicroDigital Co., Ltd., Seongnam, Republic of Korea) with six replicates. The standard curve with gallic acid (0–200 nmol) was used to estimate nanomoles of phenolic compounds (gallic acid equivalents) in the samples.

In the case of TFC estimation, a modified version of the NaNO_2_-AlCl_3_ method [41] was used. The methanol extract (60 µL) was combined with 680 µL of 30% methanol–water and 30 µL of 0.5 M NaNO_2_, vortexed for 20 s, and incubated in the dark at RT for 3 min. Next, 30 µL of 0.3 M AlCl_3_·6H_2_O was added, vortexed, incubated for another 3 min and followed by 200 µL of 1 M NaOH. After 40 min in the dark at RT, the mixture was centrifuged (5000× *g*, 1 min). Absorbance was measured at 506 nm with six replicates. TFC was quantified using rutin as a standard.

### 4.6. The Ascorbate/Dehydroascorbate (AsA/DAsA) Ratio

AsA and DAsA were quantified using the bipyridyl method [42], which measures Fe^3+^ reduction by AsA. Samples were pre-incubated with dithiothreitol (DTT) to reduce DAsA to AsA, and excess DTT was neutralized with N-ethylmaleimide. Absorbance was recorded at 525 nm. Calibration curves were prepared with L-ascorbic acid (0–1 µM). For extraction, 500 mg of plant tissue was homogenized in 1.5 mL of 6% TCA, centrifuged (15,000× *g*, 4 °C, 5 min), and the supernatant was analyzed immediately. Each treatment was tested in six replicates.

### 4.7. Anthocyanin (ANT) Assay

The ANT content was measured as described previously [43]. Plant tissue (200 mg) was extracted with 1 mL methanol–HCl (99:1, *v*/*v*) at 4 °C. Absorbance was measured at 530 and 657 nm, and relative ANT levels [AU g^−1^ FW] were calculated using Equation (1). Six replicates were analyzed for each treatment.(1)$$\frac{{Abs}_{530}-\left(0.25\times {Abs}_{657}\right)\times extraction\;volume\;\left(mL\right)\times 1}{Mass\;of\;tissue\;sample\;\left(g\right)}=\frac{Relative\;units\;of\;anthocyanins\;}{g\;Fresh\;weight\;of\;plant\;tissue\;}$$

### 4.8. Antioxidant Activity by DPPH Assay

DPPH radical scavenging activity was assessed using 60 µL of methanol extract, mixed with 904 µL methanol and 576 µL 0.125 mM DPPH in methanol, vortexed for 20 s, and incubated at 37 °C for 30 min. Absorbance was measured at 517 nm with six replicates. Radical scavenging activity was calculated using Equation (2) and a BHT calibration curve (0–400 µg/mL) [44].(2)$$DPPH\;inhibition\;\left(\%\right)=\frac{Absorbance\;of\;control*-Absrobance\;of\;sample}{Absorbance\;of\;control}\times 100$$

* Control indicates DPPH mixture incubated with 0 µg BHT solution.

### 4.9. Lipid Peroxidation

Lipid peroxidation was quantified via the malondialdehyde (MDA) content using a thiobarbituric acid (TBA) assay [45]. Leaf tissue (200 mg) was homogenized in 1 mL of methanol, incubated at 60 °C for 30 min, and centrifuged (10,000× *g*, 5 min). Extracts were mixed with TCA-BHT-TBA solution, incubated at 95 °C for 5 min, and centrifuged again. Absorbance was measured at 532, 600, and 450 nm, and MDA was calculated using Equation (3).(3)$$MDA=6.45\times \left({Abs}_{532}-{Abs}_{600}\right)-0.56\times {Abs}_{450}$$

### 4.10. Data Fitting and Statistical Analysis

DPPH inhibition data were fitted using OriginPro 2024b (OriginLab Corporation, Northampton, MA, USA). Statistical analyses were conducted using Statistica 13.3 (StatSoft Inc., Oklahoma, OK, USA). Normality was verified with the Shapiro–Wilk test, and variance equality was verified with Levene’s test. One-way ANOVA and Tukey’s HSD test identified significant differences (*p* < 0.05). Pearson’s correlations were calculated in OriginPro v. 2024b, with significance thresholds set at * *p* = 0.05, ** *p* = 0.01, and *** *p* = 0.001.

## 5. Conclusions

This study demonstrates that the preharvest supplementation of an RGB light spectrum with low-dose UV-B or UV-C differentially modulates antioxidant capacity, secondary metabolite accumulation, and photosynthetic integrity in basil microgreens. Firstly, a four-day UV-B exposure significantly increased the total flavonoid content (TFC) and anthocyanins (ANT) in the green-leaf basil (SL) but reduced phenolic compound accumulation in the purple-leaf cultivar (DO). The different responses of the analyzed cultivars to UV-B were presumably related to the significantly higher initial accumulation of UV-screening pigments in DO. Consequently, despite the fact that the UV-B treatment promoted UVR8 monomerization in both cultivars, the enhanced CHS level and SMs synthesis was noted only for the SL basil. Thus, we speculate that the attenuation of UVR8 downstream signaling might have been due to elevated HY5 activity or RUP-mediated dimerization of UVR8. Secondly, the results show that the UV-C treatment was also effective in enhancing the total phenolic content (TPC) and ANT level in SL as a result of CHS activation this time, albeit in a UVR8-independent but ROS-related manner. Conversely, in DO, UV-C decreased TPC and AsA, leading to reduced total antioxidant capacity (TAC). Thirdly, while the UV-B treatment preserved photosynthetic efficiency and improved plant morphology and leaf biomass accumulation, UV-C exposure induced oxidative stress and the disruption of photosystem integrity, with SL exhibiting greater susceptibility than DO. Moreover, both cultivars showed negative morphological and biomass traits after UV-C light exposure.

These findings highlight cultivar-specific UV responses and suggest that low-dose UV-B is a viable strategy for enhancing antioxidant synthesis in basil without compromising plant quality, whereas UV-C is unsuitable due to adverse morphological effects.

## Figures and Tables

**Figure 1 ijms-26-02352-f001:**
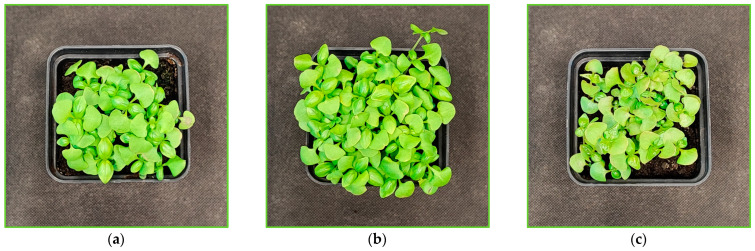
Morphology of microgreen basil (cv. Sweet Large) at 20 DAS under (**a**) RGB (control), (**b**) RGB+UV-B (311 nm), and (**c**) RGB+UV-C (254 nm).

**Figure 2 ijms-26-02352-f002:**
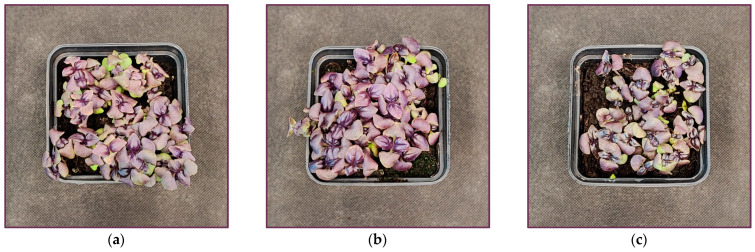
Morphology of microgreen basil (cv. Dark Opal) at 20 DAS under (**a**) RGB (control), (**b**) RGB+UV-B (311 nm), and (**c**) RGB+UV-C (254 nm).

**Figure 3 ijms-26-02352-f003:**
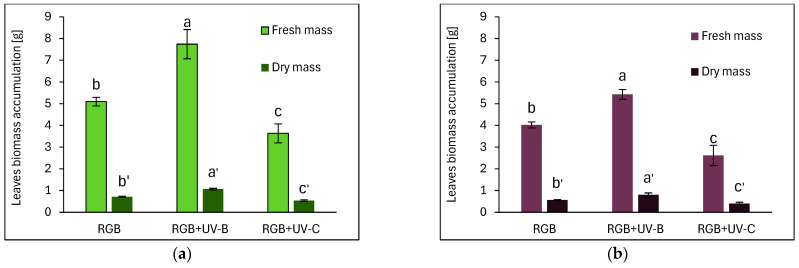
Leaf biomass accumulation of microgreen basil (**a**) cv. Sweet Large and (**b**) cv. Dark Opal at 20 DAS under RGB (control), RGB+UV-B (311 nm), or RGB+UV-C (254 nm). Leaf biomass was assessed as a fresh mass (FM) or dry mass (DM) and relates to the mass of all leaves of individual plants (with petioles). Data represent means ± SD (*n* = 10). Different letters (a–c or a’–c’) indicate significant differences between treatments (*p* ≤ 0.05, Tukey’s HSD test).

**Figure 4 ijms-26-02352-f004:**
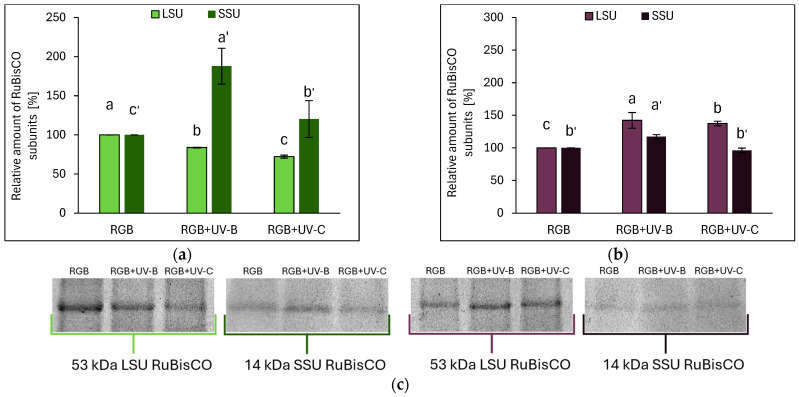
Densitometric analysis of RuBisCO large (LSU) and small (SSU) subunit of control (RGB), UV-B treated (RGB+UV-B), or UV-C treated (RGB+UV-C) plants of microgreen basil (*Ocimum basilicum* L.) cultivar with (**a**) green leaf (cv. Sweet Large) or (**b**) purple leaf (cv. Dark Opal) after short-term (1–4 day) progressive exposition to UV light at 20 DAS (days after sowing). Beneath (**c**) the LSU (53 kDa) or SSU (14 kDa) protein bands of leaf proteins resolved in a 4–20% TGX polyacrylamide gel and visualized with Coomassie Blue. The relative amounts (%) of RuBisCO subunits were normalized to RGB control. Data represent means ± SD (*n* = 3). Different letters (a–c for LSU or a′–c′ for SSU) indicate significant differences between treatments (*p* ≤ 0.05, Tukey’s HSD test).

**Figure 5 ijms-26-02352-f005:**
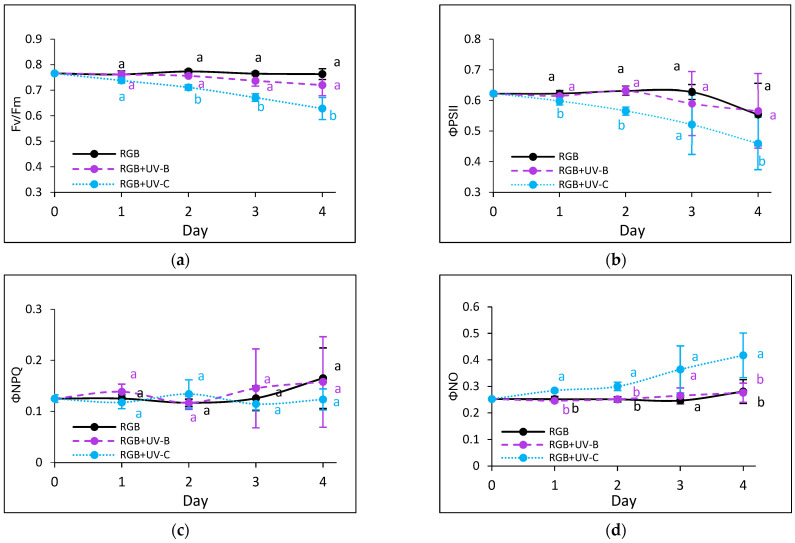
Chlorophyll *a* fluorescence in control (RGB), UV-B-treated (RGB+UV-B), and UV-C-treated (RGB+UV-C) microgreen basil (cv. Sweet Large) after 1–4 days of progressive UV exposure at 20 DAS, including (**a**) Fv/Fm (maximum quantum yield of PSII), (**b**) ΦPSII (effective PSII quantum yield), (**c**) ΦNPQ (regulated energy dissipation), (**d**) ΦNO (non-regulated dissipation), (**e**) NPQ (non-photochemical quenching), and (**f**) ETR (electron transport rate). Data represent means ± SD (*n* = 6). Different letters (a, b) indicate significant differences (*p* ≤ 0.05, Tukey’s HSD test).

**Figure 6 ijms-26-02352-f006:**
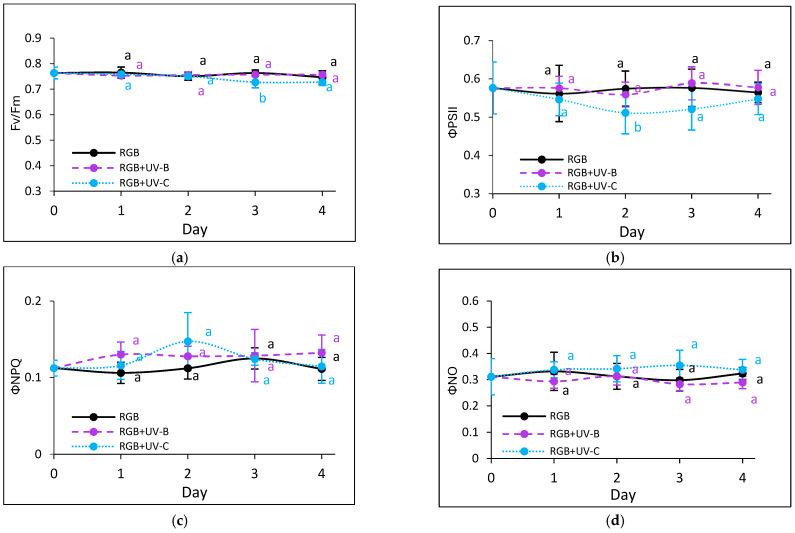
Chlorophyll *a* fluorescence in control (RGB), UV-B-treated (RGB+UV-B), and UV-C-treated (RGB+UV-C) microgreen basil (cv. Dark Opal) after 1–4 days of progressive UV exposure at 20 DAS, including (**a**) Fv/Fm (maximum quantum yield of PSII), (**b**) ΦPSII (effective PSII quantum yield), (**c**) ΦNPQ (regulated energy dissipation), (**d**) ΦNO (non-regulated dissipation), (**e**) NPQ (non-photochemical quenching), and (**f**) ETR (electron transport rate). Data represent means ± SD (*n* = 6). Different letters (a, b) indicate significant differences (*p* ≤ 0.05, Tukey’s HSD test).

**Figure 7 ijms-26-02352-f007:**
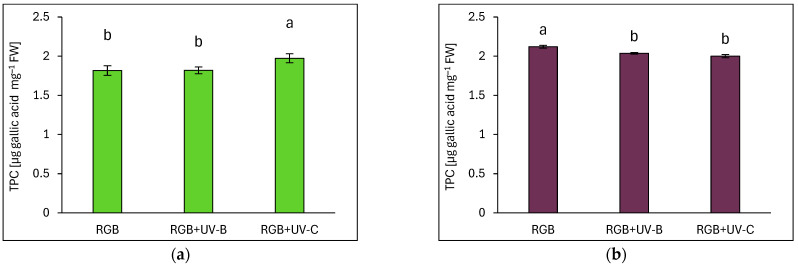
Total phenolic content (TPC) of control (RGB), UV-B-treated (RGB+UV-B), and UV-C-treated (RGB+UV-C) microgreen basil (*Ocimum basilicum* L.) cultivars with (**a**) green leaves (cv. Sweet Large) and (**b**) purple leaves (cv. Dark Opal) at 20 DAS (days after sowing), expressed as µg gallic acid equivalents per mg of fresh weight (FW). Bars represent means ± SD (*n* = 6). Different letters (a, b) indicate significant differences (*p* ≤ 0.05, Tukey’s HSD test).

**Figure 8 ijms-26-02352-f008:**
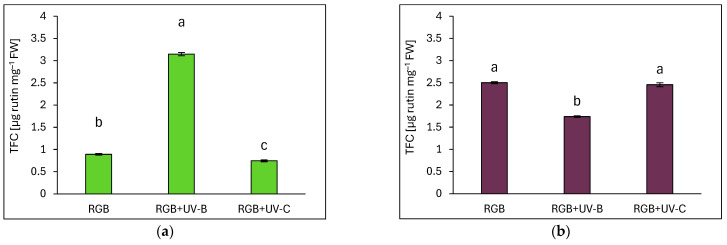
Total flavonoid content (TFC) of control (RGB), UV-B-treated (RGB+UV-B), and UV-C-treated (RGB+UV-C) microgreen basil cultivars with (**a**) green (cv. Sweet Large) and (**b**) purple leaves (cv. Dark Opal) at 20 DAS, expressed as µg rutin equivalents per mg of FW. Bars show means ± SD (*n* = 6). Different letters (a–c) denote significant differences (*p* ≤ 0.05, Tukey’s HSD test).

**Figure 9 ijms-26-02352-f009:**
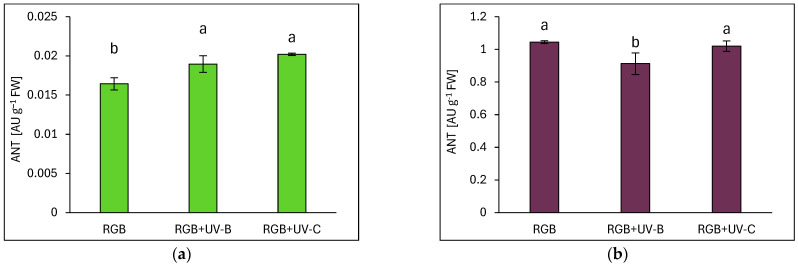
Anthocyanin (ANT) concentrations in control (RGB), UV-B-treated (RGB+UV-B), and UV-C-treated (RGB+UV-C) microgreen basil cultivars with (**a**) green (cv. Sweet Large) and (**b**) purple leaves (cv. Dark Opal) at 20 DAS, measured as arbitrary units (AUs) per g of FW. Bars represent means ± SD (*n* = 6). Different letters (a, b) indicate significant differences (*p* ≤ 0.05, Tukey’s HSD test).

**Figure 10 ijms-26-02352-f010:**
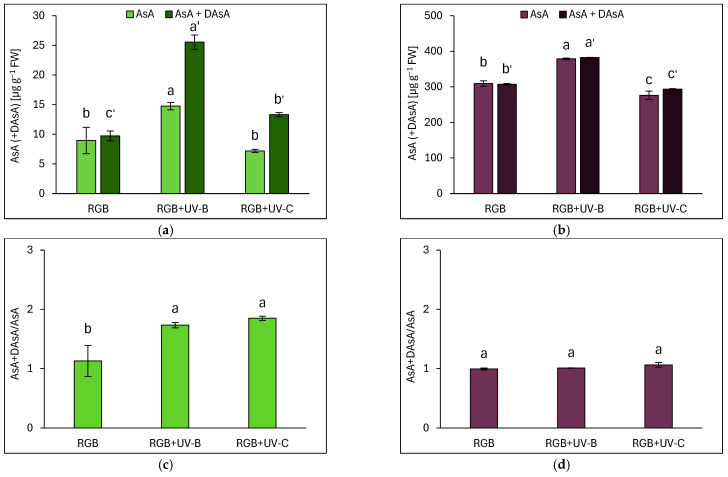
Initial ascorbic acid (AsA), total AsA pool (AsA+DAsA), and AsA+DAsA/AsA ratio in control (RGB), UV-B-treated (RGB+UV-B), and UV-C-treated (RGB+UV-C) microgreen basil cultivars with (**a**,**c**) green (cv. Sweet Large) and (**b**,**d**) purple leaves (cv. Dark Opal) at 20 DAS. Initial AsA was measured directly using the bipyridyl method, while total AsA was quantified after reducing dehydroascorbic acid (DAsA) with dithiothreitol (DTT). Bars show means ± SD (*n* = 6). Different letters (a–c or a′–c′) indicate significant differences (*p* ≤ 0.05, Tukey’s HSD test).

**Figure 11 ijms-26-02352-f011:**
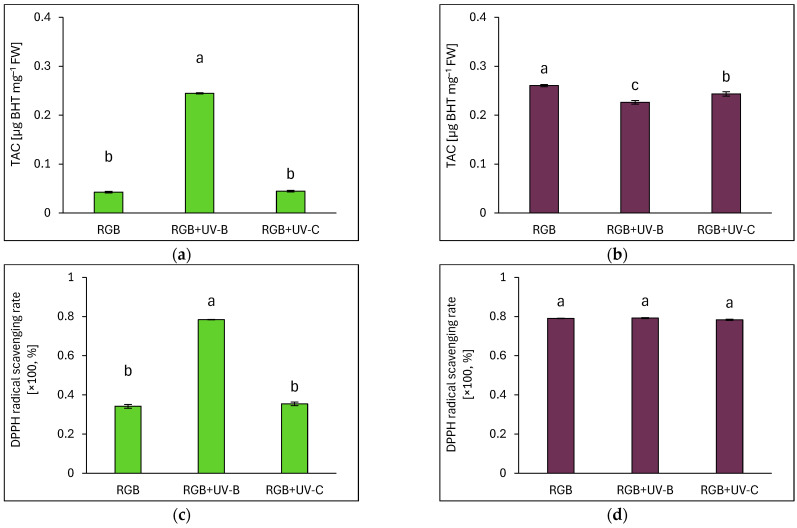
Total antioxidant capacity (TAC) and DPPH radical scavenging activity in control (RGB), UV-B-treated (RGB+UV-B), and UV-C-treated (RGB+UV-C) l basil cultivars with (**a**,**c**) green (cv. Sweet Large) and (**b**,**d**) purple leaves (cv. Dark Opal) at 20 DAS, expressed as µg BHT equivalents per mg of FW. Bars represent means ± SD (*n* = 6). Different letters (a–c) denote significant differences (*p* ≤ 0.05, Tukey’s HSD test).

**Figure 12 ijms-26-02352-f012:**
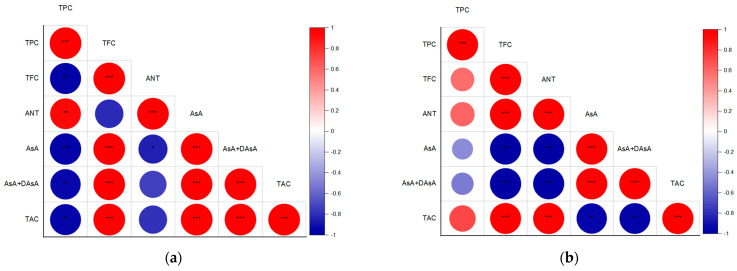
Heatmap showing Pearson’s correlations among TPC, TFC, ANT, initial AsA, total AsA pool, and TAC in control (RGB), UV-B-treated (RGB+UV-B), and UV-C-treated (RGB+UV-C) microgreens basil cultivars with (**a**) green (cv. Sweet Large) and (**b**) purple leaves (cv. Dark Opal) at 20 DAS. Positive correlations are shown in red and negative correlations in blue. Circle size reflects correlation coefficients. Asterisks (*, **, ***) indicate significance at *p* ≤ 0.05, 0.01, or 0.001, respectively.

**Figure 13 ijms-26-02352-f013:**
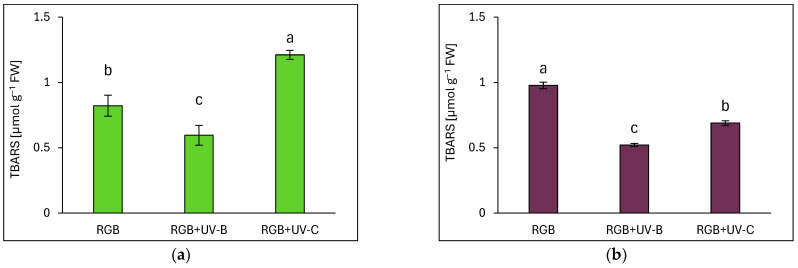
Lipid peroxidation rates assessed by TBARS levels in control (RGB), UV-B-treated (RGB+UV-B), and UV-C-treated (RGB+UV-C) microgreen basil cultivars with (**a**) green (cv. Sweet Large) and (**b**) purple leaves (cv. Dark Opal) at 20 DAS. Bars represent means ± SD (*n* = 6). Different letters (a–c) indicate significant differences (*p* ≤ 0.05, Tukey’s HSD test).

**Figure 14 ijms-26-02352-f014:**
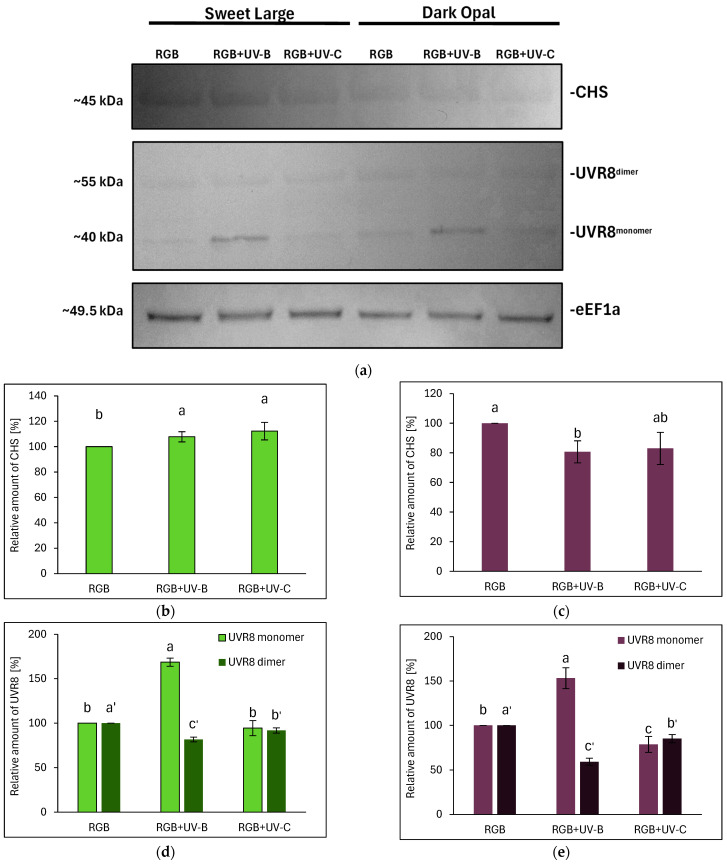
Western blot analyses of chalcone synthase (CHS), ultraviolet-B receptor UVR8 (monomer and homodimer), and elongation factor 1-alpha (eEF1a, loading control) in fully expanded basil leaves of green (**a**,**b**,**d**) and purple (**a**,**c**,**e**) cultivars at 20 DAS. Proteins (5 μg/lane) were resolved on 4–20% TGX stain-free polyacrylamide gels, transferred to nitrocellulose membranes, and visualized with DAB (**a**). Quantifications (**b**–**e**) were performed using ImageJ. Bars represent means ± SD (*n* = 3). Different letters (a–c or a′–c′) indicate significant differences (*p* ≤ 0.05, Tukey’s HSD test).

**Figure 15 ijms-26-02352-f015:**
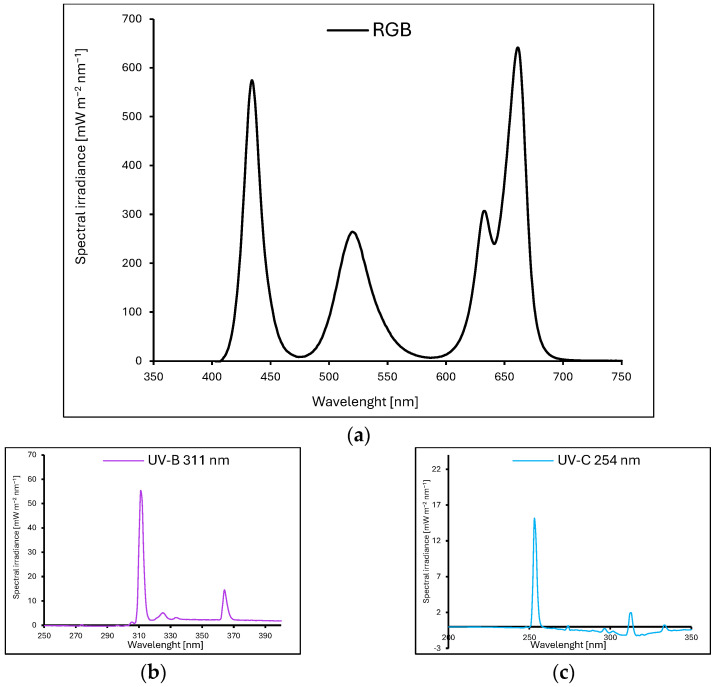
Light spectra of lamps were recorded at four locations, averaged for (**a**) RGB-only (control), (**b**) UV-B (311 nm), and (**c**) UV-C (254 nm). All plants were grown under a 200 µmol m^−2^ s^−1^ RGB spectrum treatment for 20 days, with UV supplementation applied during the final 4 days before harvest.

**Table 1 ijms-26-02352-t001:** Photosynthetic pigments and soluble protein content (SLP) in microgreen basil cv. Sweet Large under varying light conditions.

Parameter	Treatment		
	RGB	RGB+UV-B	RGB+UV-C
Chlorophyll *a + b* [mg g^−1^ FW]	1.405 ± 0.013 ^a^	0.878 ± 0.006 ^b^	1.423 ± 0.005 ^a^
Chlorophyll *a* [mg g^−1^ FW]	1.000 ± 0.010 ^a^	0.617 ± 0.004 ^c^	0.991 ± 0.004 ^b^
Chlorophyll *b* [mg g^−1^ FW]	0.405 ± 0.003 ^b^	0.261 ± 0.002 ^c^	0.432 ± 0.002 ^a^
Chlorophyll *a*/*b*	2.470 ± 0.006 ^a^	2.364 ± 0.011 ^b^	2.295 ± 0.012 ^c^
Carotenoids [mg g^−1^ FW]	0.161 ± 0.001 ^a^	0.113 ± 0.000 ^c^	0.144 ± 0.001 ^b^
Chlorophyll *a + b*/carotenoids	8.732 ± 0.006 ^b^	7.799 ± 0.034 ^c^	9.905 ± 0.054 ^a^
Soluble leaf proteins [mg g^−1^ FW]	60.32 ± 0.83 ^c^	74.52 ± 0.33 ^b^	78.09 ± 0.90 ^a^

The presented values are means of six (or four for SLP) replicates ± SD. Different letters (a–c) in the same row indicate significant differences between treatments at *p* ≤ 0.05 with a Tukey’s HSD test. FW—fresh weight.

**Table 2 ijms-26-02352-t002:** Photosynthetic pigments and SLP in microgreen basil cv. Dark Opal under varying light conditions.

Parameter	Treatment		
	RGB	RGB+UV-B	RGB+UV-C
Chlorophyll *a + b* [mg g^−1^ FW]	1.384 ± 0.008 ^b^	0.786 ± 0.003 ^c^	1.419 ± 0.007 ^a^
Chlorophyll *a* [mg g^−1^ FW]	0.939 ± 0.006 ^b^	0.526 ± 0.002 ^c^	0.966 ± 0.005 ^a^
Chlorophyll *b* [mg g^−1^ FW]	0.444 ± 0.003 ^b^	0.261 ± 0.001 ^c^	0.453 ± 0.003 ^a^
Chlorophyll *a*/*b*	2.115 ± 0.003 ^b^	2.015 ± 0.005 ^c^	2.132 ± 0.009 ^a^
Carotenoids [mg g^−1^ FW]	0.181 ± 0.001 ^b^	0.089 ± 0.001 ^c^	0.188 ± 0.001 ^a^
Chlorophyll *a + b*/carotenoids	7.646 ± 0.026 ^b^	8.805 ± 0.025 ^a^	7.553 ± 0.037 ^c^
Soluble leaf proteins [mg g^−1^ FW]	127.74 ± 0.67 ^a^	109.36 ± 0.22 ^b^	106.26 ± 0.86 ^c^

The presented values are means of six (or four for SLP) replicates ± SD. Different letters (a–c) in the same row indicate significant differences between treatments at *p* ≤ 0.05 with a Tukey’s HSD test. FW—fresh weight.

**Table 3 ijms-26-02352-t003:** The schedule of supplemental UV-B and UV-C light treatment.

Treatment, Wavelength Peak (nm)	Daily Time Exposure (min), Diurnal Time				Total Time (h)	Total Irradiance (W m^−2^)	Irradiance (PAR) (W m^−2^)	Peak-to-Background Irradiance Ratio *	Cumulative Dose (kJ m^−2^)
	Day 1	Day 2	Day 3	Day 4					
UV-B, 311	15 12.00–12.15 p.m.	30 12.00–12.30 p.m.	60 12.00–1.00 p.m.	120 12.00–2.00 p.m.	3.75	1.1572	0.253	5.32	15.622
UV-C, 254	7.5 12.00–12.08 p.m.	15 12.00–12.15 p.m.	30 12.00–12.30 p.m.	60 12.00–1.00 p.m.	1.875	0.8901	0.177	10.1	6.008

UV-B indicates ultraviolet B light; UV-C indicates ultraviolet C light; PAR indicates photosynthetically active radiation. * For UV-B, the assessed ratio relates to an irradiance integral of 311 ± 5 nm to 365 ± 5 nm. For UV-C, the ratio relates to an irradiance integral of 254 ± 5 nm to 311 ± 5 nm.

## Data Availability

The data presented in this study are available on request from the corresponding author. The data are not publicly available due to the strict management of various data and technical resources within the research teams.

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
