# Peer review of "Enhancing the Quality of Indoor-Grown Basil Microgreens with Low-Dose UV-B or UV-C Light Supplementation"

_ijms, 2025, doi:10.3390/ijms26052352_

Round 1
Reviewer 1 Report
Comments and Suggestions for Authors
The current manuscript investigated the effects of UVB and UVC on quality formation among different basil cultivars, results revealed that different active compounds displayed distinct accumulation patterns between radiation treatments and cultivars, microcosmic influences were also recorded. The results are of importance for the light environment construction and quality control in indoor farming, also help researchers to understand the different function mechanisms of UVB and UVC. Besides, several suggestions questions are raised as follow:
1) The presentation order of results can be optimized. For example, the morphological results can be moved to the front of Results, then physiological data, metabolic data;
2) Biomass between treatments exhibited significant difference while the specific data was not shown, this is also meaningful for indoor farming, please complete it;
3) In the Light Treatments, the daily exposure time was increased in the following days, please explain it;
4) In the Spectral curve (Fig 14 b,c), there exists a obvious peak along with the primary spectrum, will it disturb the final results.
Reviewer 2 Report
Comments and Suggestions for Authors
- Please provide more recent references for the Introduction, especially for greens such as basil or similar plants.
- Provide additional details regarding any experimental design that was implemented, including number of pots and plants per pot, if applicable.
- Define DPPH in line 481
